# Contribution of Home Gardens to Sustainable Development: Perspectives from A Supported Opinion Essay

**DOI:** 10.3390/ijerph192013715

**Published:** 2022-10-21

**Authors:** Mário Santos, Helena Moreira, João Alexandre Cabral, Ronaldo Gabriel, Andreia Teixeira, Rita Bastos, Alfredo Aires

**Affiliations:** 1Laboratory of Fluvial and Terrestrial Ecology, Innovation and Development Center, University of Trás-os-Montes e Alto Douro, 5000-911 Vila Real, Portugal; 2Laboratory of Ecology and Conservation, Federal Institute of Education, Science and Technology of Maranhão, Rua do Comercio, 100, Buriticupu 65393-000, MA, Brazil; 3CITAB—Centre for the Research and Technology of Agro-Environment and Biological Sciences, Institute for Innovation, Capacity Building and Sustainability of Agri-Food Production (Inov4Agro) and Department of Biology and Environment, University of Trás-os-Montes e Alto Douro, 5000-801 Vila Real, Portugal; 4Department of Sports, Exercise and Health Sciences, University of Trás-os-Montes e Alto Douro, 5000-911 Vila Real, Portugal; 5CIDESD—Research Center in Sports Sciences, Health Sciences and Human Development, University of Trás-os-Montes e Alto Douro, 5000-801 Vila Real, Portugal; 6CIBIO/InBIO/BioPolis, Centro de Investigação em Biodiversidade e Recursos Genéticos, Universidade do Porto, 4485-661 Vairão, Portugal; 7Department of Agronomy, School of Agrarian and Veterinary Sciences, University of Trás-os-Montes e Alto Douro, UTAD, Quinta de Prados, 5000-801 Vila Real, Portugal

**Keywords:** home gardening, health benefits, human wellbeing, food security, food quality, biodiversity, sustainability, ecosystem services

## Abstract

Home gardening has a long history that started when humans became sedentary, being traditionally considered an accessible source of food and medicinal plants to treat common illnesses. With trends towards urbanization and industrialization, particularly in the post-World War II period, the importance of home gardens as important spaces for growing food and medicinal plants reduced and they began to be increasingly seen as decorative and leisure spaces. However, the growing awareness of the negative impacts of agricultural intensification and urbanization for human health, food quality, ecosystem resilience, and biodiversity conservation motivated the emergence of new approaches concerning home gardens. Societies began to question the potential of nearby green infrastructures to human wellbeing, food provisioning, and the conservation of traditional varieties, as well as providers of important services, such as ecological corridors for wild species and carbon sinks. In this context. and to foster adaptive and resilient social–ecological systems, our supported viewpoint intends to be more than an exhaustive set of perceptions, but a reflection of ideas about the important contribution of home gardens to sustainable development. We envision these humble spaces strengthening social and ecological components, by providing a set of diversified and intermingled goods and services for an increasingly urban population.

## 1. Introduction

Sustainable development has emerged as a concept associated with the growing awareness of the need to balance social–economic development with a healthy environment [1]. Moreover, sustainable development goals (SDGs) have evolved to not only include the people, planet, and prosperity, but also peace and partnership, linking the economic, social, and environmental dimensions of sustainability [2]. Even though SDGs are not simple to be applied considering the possible incompatibilities and divergent points of view among social players, they have become a fundamental component of policy frameworks by diverse stakeholders “sitting at the same table”, such as governmental agencies, researchers, civil society, and the private sector (among others), to discuss their implementation [3]. This has contributed to the vision of a global human development approach, where diverse indicators have been considered as complementary and mutually reinforcing each other [4]. Moreover, global or regional SDG frameworks can only be effective when completed locally, considering the specificity of socio-ecological systems [5]. Additionally, this implementation is supported by resilience, a key feature of sustainable social–ecological systems [6], demonstrated by their capability of reorganization after disturbance [7] and by the amount of ecosystem services that translate into contributions of nature to human wellbeing and health [8].

Given the demographic shift towards urbanization, cities are likely to increase in importance and will be the main grounds for the SDGs’ implementation [9]. In fact, 54% of the world’s population lives in urban areas—a number that is expected to increase to approximately 59% by 2030. Thus, to meet SDGs, governments should tackle the direct and remote impacts of their growing cities [10]. Therefore, urban and suburban green spaces, encompassing public gardens, urban forests, and private backyards and home (family) gardens, but also rural home gardens in monocultural/industrial agricultural landscapes, should be considered as fundamental infrastructures in the road to sustainable societies [11]. These areas could provide several types of increasingly uncommon but essential ecosystem services [12]. For example, urban, suburban, and rural gardens, and especially the network of home gardens, are significant land uses, surpassing in many countries the area occupied by commercial crops and natural habitats [13]. On the other hand, rural home gardens are not well represented by traditional mapping approaches, but studies confirm the spatial importance of rural home gardens in several countries, from tropical to temperate regions [14,15,16,17,18]. These rural home gardens are in relapse, jeopardized by abandonment and by the increasing area occupied by intensive large-scale agriculture [19,20,21]. Even so, home gardens, generally defined as a non-built land portions close to the household (more information in Section 1.1.1), are usually considered of minor importance within socioecological systems, perhaps due to their average small size, non-commercial use, and “unregulated” management [22,23]. Additionally, several works demonstrate their importance in the provision of a range of social, economic, and environmental services (e.g., [24,25,26,27]). Food production, income support, physical activity, wellbeing, and a connection with nature were all highlighted in diverse works (e.g., [28,29,30]). Additionally, the net of home gardens in a region might contribute to improve resilience within disruptive scenarios by reinforcing social equity and justice [31,32]. Some studies have also linked these small areas with biodiversity and nature conservation, even if complex multi-factorial factors with direct and indirect impacts on species diversity and abundance are yet to be fully understood [25,33,34]. In this way, the main objective of this work is to discuss what the authors have considered significant features of home gardens (and home gardening) associated with the SDGs, paying special attention to wellbeing and health, nutrition and carbon footprint reduction, and biodiversity and nature conservation (Appendix A). With this in mind, the authors suggest looking with eyes wide open at the vast possibilities of home gardens to tackle sustainable development goals and include in the discussion conceptual ideas that might help to enlighten their overall value.

### 1.1. Home Garden Definition and Methodological Framework

#### 1.1.1. Home Garden Definition

A garden is a planned space, usually outdoors, set aside for the cultivation, display, and enjoyment of plants and other forms of nature [13]. Within gardens, home gardens are small areas (usually much less 1 hectare) surrounding the residential parcel, usually associated with family use (most home gardens are family gardens), characterized by combinations of various perennial and annual plants, sometimes in association with domestic animals and might include additional infrastructures, such as ponds, greenhouses, and green roofs [13]. From purely aesthetic gardens to food production spaces, several gardens include mixed areas (e.g., agroforests) and have diverse uses. In fact, home garden “architecture” and organization, the species chosen, and the management options are linked with the local ecological conditions, but mostly with the options of the members of the household, providing a diverse and stable supply of services and benefits to families [35,36]. Even if they are associated with urban domestic/family gardens and/or self-consumption, home gardens are also an important land-use component in peri-urban and rural areas and in local food markets [37].

#### 1.1.2. Supported Opinion Methodological Framework

A pre-opinion online and face-to-face forum occurred during the 2nd semester in 2021, involving three co-authors of the manuscript, in order to debate the personal views and importance of home gardens to tackle several of the sustainable development goals. For the purposes of the opinion manuscript, as each one of the co-authors’ expertise focused on different scientific domains, namely, healthy lifestyles and green exercise (Helena Moreira); food production and food quality (Alfredo Aires); and agroecosystems, biodiversity, and conservation ecology (Mário Santos), the idea of the preliminary discussions was to compare and define a common view concerning home gardens and to guide the workflow associated with this [38]. An expert draft was produced with his/her viewpoint, sharing and confronting with the other co-authors and justifying, whenever possible, their expert opinion using the relevant bibliographic information. Subsequently, during the 1st semester in 2022, the forum was enlarged to include additional researchers’ opinions (Andreia Teixeira, João Alexandre Cabral, Rita Bastos and Ronaldo Gabriel), comments, and discussion of ideas, obtaining a consensus regarding several issues [39]. This former step enabled the introduction of complementary remarks and risks that were integrated in the discussion. The final manuscript highlighted the key themes and perceptions that emerged during the discussions, based on the sound evidence and research, usually termed a supported opinion essay (Figure 1).

### 1.2. An Appraisal of Health and Wellbeing Contributions from Home Gardens and Home Gardening Activities

Several of the issues below are linked to home gardens and gardening, even if they should also be considered with other types of “nature immersion” and outdoor activities. Direct visual contact with flowers, green plants, and wood has positive effects on brain activity, decreasing sympathetic stimulation and increasing parasympathetic activity [40]. Additionally, visual and olfactory stimulation generated by the presence of leafy plants or fresh flowers decreases oxyhemoglobin concentration (a form of hemoglobin that carries oxygen) in the right, prefrontal cortex, generating a physiological-relaxation effect [41,42,43]. In fact, the anterior part of the frontal lobe of the brain is specialized in affective processing, aggregating information from the sensory cortexes, brainstem, and amygdala, and regulating hyperactivity in depressed people [43]. Additionally, plant and animal diversity are associated with increased attention restoration, with implications in the reduction in stress levels [44]. According to [45], the practice of home gardening (and gardening in general) (i.e., at least 1 to 4 h per week) is reflected in greater human resilience, influencing factors, such as emotional regulation, interrelatedness, confidence, positive thinking, and spirituality. These relationships, particularly evident in older people, are mediated by restored attention, increased physical activity, and self-esteem, fascination (the gardener’s motivation to seek knowledge and exchange ideas with others), and identity with the home garden (a sense of satisfaction and accomplishment in planning, planting, and harvesting what he or she has sown), and the encouragement of socialization.

The individual’s greater ability to adapt to stressful situations and deal with adverse conditions and unexpected changes is particularly evident in home (and community) gardening [46,47], which promotes contact with nature in a sustainable way while reducing nutrition-based health inequalities, particularly relevant in the elderly, refugees [47], and ethnic and racial minorities [48]. Table 1 illustrates the intensity classification of various gardening tasks in metabolic equivalents (1 MET = 3.5 mL/kg/min), mirroring the energy expenditure associated with performing such tasks and classifying them into three intensity levels: light (<3 METs), moderate (3–5.9 METs), and vigorous (≥6 METs) [35]. An individual weighing 70 kg engaged in planting trees in his garden (4.0 METs), for example, will expend 157.5 kcal in about 1/2 an hour (4.0 METs * 70 kg * (30 min)/(60 min)), depending on factors, such as fitness level, gender, and environmental conditions [49,50].

Carrying the wheelbarrow, pulling weeds, twisting, and bending while planting, among others, are some of the tasks that contribute to the improvement of strength, balance, and flexibility, leveraging their diversity for reducing the risk of injury and relieving the fatigue related with the repeated actions [50].

A study involving middle-aged (and older adults) gardeners revealed that those who were physically active (≥150 min of moderate-intensity physical activity per week) exhibited better physical health and handgrip strength, compared to those who gardened between 120 and 149 min/week, and especially to those who did so for less than 120 min/week [51]. According to a number of authors, people are more likely to document improvements to wellbeing and health when exposure to nature has a minimum duration of 120 min per week [52], with physical, psychological, and social benefits being magnified by spending more time on gardening activities [27].

Handgrip strength is stimulated by the variety of tasks associated with home gardening that require the release and flexion of the thumb and forefinger, and its improvement is associated with cognitive [53], oncological [54], metabolic [55], and bone [56] benefits, decreasing cardiovascular and non-cardiovascular diseases and mortality risks [57]. The effort developed when gardening and the exposure to natural light increases individuals’ alertness during the day, improving sleep duration and quality [58]. This might counteract socio-environmental and lifestyle factors, such as stress, temperature, humidity, and work-shift influence regarding the production of melatonin, i.e., sleep and wake cycles [59]. In fact, this hormone, produced by the pineal gland in the absence of light stimuli, regulates sleep and contributes to antioxidant and anti-inflammatory properties that protect the body from various diseases, such as cancer, diabetes, and metabolic syndrome [60].

For older people (and particularly women), especially affected by insomnia (and overweight) that is related with anxiety, depression [61], and reduced levels of physical activity, home gardening may help to improve sleep disorders. Through the exposure of the skin to ultraviolet radiation, this outdoor activity also stimulates vitamin D synthesis, increasing levels that are usually reduced in middle-aged and elderly individuals [62], as well as waiving the use of supplements [63]. The health benefits of vitamin D have been recognized, including their role in regulating glucose metabolism, decreasing cardiovascular diseases [62], improving cognitive ability [64], preventing depression [65] and some types of cancer [63,66], osteoporosis, multiple sclerosis, and COVID-19 [63]. Observational studies have also identified an association of vitamin D levels with arthritis, chronic pain [67], and low-back pain [68].

Plant diversity associated with home gardens (and in general, plant diverse systems) increases the microbiome, with potential effects in mitigating the acute and chronic health effects of air pollution, including allergies, asthma, and chronic inflammatory diseases [69]. Planting, digging, weeding, or consuming home-garden-grown products, including fruits and vegetables, also increases the gut microbiome and induces a higher intake of fiber, iron, selenium, and vitamins C and K [70], due the presence of vitamin-synthetizing bacterium in the soil (e.g., *Mycobacterium vaccae*) [71]. Studies using mice have revealed that the bacterium is active in a specific set of serotonin-producing neurons located in the subregion of the dorsal raphe nucleus (neuronal aggregates divided into pairs along the brainstem), and that it stabilizes the gut microbiome, improving the response to stimuli that triggers stress and anxiety. Their effects in protecting allergic bronchial asthma [72] and on the response to chemotherapy in some types of cancer were also documented in the literature [73]. Exposure to the environmental microbiome and other elements of nature, including phytocides (volatile, antimicrobial, organic compounds emitted as a defense mechanism by plants), negative air ions, sunlight, and sights and sounds also provide analgesia, and these benefits are enhanced with exposure to biodiverse spaces [74].

Additionally, light-to-moderate-intensity gardening activities that might occur near the household are associated with cognitive health benefits, namely, by an upsurge in the brain-derived neurotrophic factor (BNDF) [75,76] and platelet-derived growth factor (PDGF) [75]. Both growth factors are related to memory and cognitive function; their levels decrease with age, implying a reduction in brain volume and weight (5% per decade after the 4th decade, especially relevant after the age of 70 years) [77]. The BNDF, associated with neurogenesis, synaptic transmission, and production of tryptophan, an amino acid precursor of serotonin, is linked with the hippocampus and cerebral cortex [76]; on the other hand, PDGF promotes cell proliferation/growth and neuronal functions [75]. Neurogenesis and the stimulation of new synapses are enhanced if gardening is practiced for at least 3 months at a moderate intensity and with sessions lasting no less than 20 min [78].

Regular gardening, promoted when the garden is near the residence (home gardens), might reduce the risk of dementia by 36% in people over 60 years of age [79], encouraging positive-mood enhancement in individuals with average-to-advanced levels of disease [80]. Sensory stimulation derived from light, smells, and touch allows people to recall meaningful memories and past skills [81], to be engaged in the accomplishment of meaningful and productive work, reinforcing the feeling of being at “home” [81,82,83]. Home gardens can also provide opportunities for people to interact with neighbors, empowering the community spirit and social connectedness, with positive reflections on mental health. Learning about the science of plants, finding innovative ways to grow them, and discovering fresh-food sources and ways to cook them are also important in preventing cognitive decline. They might also encourage the purchase of seasonal/local products and positively influence the adoption of other pro-environmental behaviors [84]. Home gardening is also an important intergenerational activity, by the sharing of skills and knowledge, stimulating recreative environments for different age groups.

### 1.3. The Contributions of Home Gardening (and Urban Agriculture) to Dietary Diversity and Carbon Footprint Reduction

Several questions could be raised when discussing the impacts of home gardening on human health and food supply. From a consumer’s point of view, home gardening is generally perceived as involving small areas surrounding houses and villages in which mixtures of flowers, potted plants, perianal bushes, and street trees, fruits, vegetables, and medicinal plants are cultivated [85,86]. Different studies have associated home gardening with a wide range of ecosystem services, such as supplying small markets with high-quality fruits and vegetables, and employment opportunities [87,88]. In fact, divergent economies (countries from North America, South America, Europe, Australia, and Asia) envision home gardens (domestic agriculture) contributing to a reduction in the world food crisis [89]. Consumers often perceive home agriculture as a supplementary strategy to assure food security, since it can be a source of income while providing direct access to a higher number of nutritionally rich foods (vegetables and fruits) [90]. An increased stability of household diets during seasonality or other temporary shortages was also pointed out [91,92,93]. These works seem to share a common observation: that home gardening can account for an important share of the local offer of perishable food items, such as vegetables or medicinal and aromatic plants, playing a vital role in the promotion of household-food self-sufficiency [91]. In addition, it seems that families involved in home agriculture have better and varied diets: several studies also reported that home gardens supplemented diets with a significant portion of proteins, vitamins, and minerals, leading to an enriched and balanced menu [29,94], while at same time sustaining crop diversity and improving a family’s resilience [95]. The production of food by families can supply up to 20–60% of their total food consumption in fresh vegetables, medicinal and aromatic plants or eggs, and even milk and meat from small animals [96], thus increasing the accessibility of affordable fresh foods and assuring a food supply during natural disasters and wars [93]. As an example, a single home garden of about 9 square meters with tomato, cucumber, musk melon, cabbage, potato, sweet potato, squash, peppers, bush peas, lettuce, spinach, kale, carrots, onions, and beets, can provide 9.2% of protein, 23% of vitamin K, 20% of vitamin C, and less amounts of other nutrients and vitamins to a household [96]. In low-income areas, the dietary deficiency of micronutrients, such as iron, zinc, iodine, and vitamin A, is more common [97], and horticultural commodities in home gardens, such as fruits and vegetables rich in minerals, fibers, and bioactive compounds (e.g., phenolics and antioxidants), partially overcome this problem, reducing malnutrition, improving food security, and increasing the availability of food [29]. According to the Food and Agriculture Organization of the United Nations [98], a high percentage of the world’s population consumes large quantities of carbohydrate-dense staples, such as maize, rice, wheat, and potatoes, which have low concentrations of essential micronutrients necessary to maintain good health and wellbeing. Therefore, vegetables and fruits provided by home gardens can be an easy way to access those micronutrients [23], particularly in isolated places or for families with a low financial budget. Home gardening can contribute to a household’s nutrition and food security by providing rapid and direct access to a diversity of foods that can be harvested, prepared, and eaten by family members on a daily basis. This is considered beneficial for a human’s nutritional status, cardiovascular health, and for reducing the probability of catching many diseases [99,100]. Dietary diversity scores have been developed as an indicator of the micronutrient adequacy of diets [101], and many studies showed that home gardening might lead to an overall increase in nutrient intake [23,102]. Studies [103,104,105] have shown that even small home gardens can provide a substantial number of micronutrients and vitamins to a household. Complementary studies have provided specific positive and descriptive evidence of home gardening impacts (in both developed and developing countries) on families’ nutrition status and diets [102,106,107,108,109]. Thus, the areas surrounding houses, often neglected, can be utilized to grow vegetables and fruits, fulfilling the nutrient requirements [109,110]. Some examples of home gardening benefits to diet are presented in Table 2.

On the other hand, even if home garden products are mostly for auto-consumption, they can also grow multiple added-value crops, including traditional medicines and home remedies for certain illnesses [29]. In the literature, it is possible to find several studies in which home and community gardens are considered inefficient, costly, and without any offer regarding a complete solution to food insecurity [111]. We can however observe the problem from another perspective: instead of considering home and community gardens as a solution to address food insecurity, we may consider them as a part of a broad answer to address a much greater issue of offering a diversity of nutritious foods, as well as the opportunity for positive health outcomes. Moreover, home gardening can be more cost-effective than buying in the stores, as gardeners can grow what they eat most and just buying the less cost-effective prod-ucts. Contribution of home (and community) gardening to food security are presented in Figure 2.

Home gardening has also been pointed out as having a positive impact on the social conditions of local populations, trough-strengthening cohesion, and the local economy [112,113]; an increase in financial revenues, the reduction in poverty risk factors have been highlighted in the literature. Another important achievement of home gardening is related to its benefits for carbon sequestration: plant cover might buffer climate change variability [114] by creating more complex canopies than modern agriculture and/or urban areas, thus ultimately modulating microclimatic conditions [115] and sequestering atmospheric carbon into the soil [116]. In addition, green-house gas (GHG) footprints of consumers through conventional agribusiness systems is far higher than of home productions [117]. Conversely, other research suggests that the impact of home gardens on the environment may depend on the specific management and cultivation methods used [118], which in turn might have negative consequences in terms of the overuse and production of fertilizers and/or the GHG emissions. Home composting from gardening waste can also produce methane and nitrous oxide, which are strong GHGs [119,120,121]. Nonetheless, GHG reductions were observed in gardening communities, compared with conventional systems, particularly when vegetable production replaced lawns [116]. In developed countries, post-harvest processes, such as storing, refrigeration, and transportation over long distances produce high GHG emissions, comparable to the production processes [122]. These postproduction emissions are considerably reduced when vegetables are grown near the places where they are consumed, as in the case of home gardens. Plants considerably reduce CO_2_ and heat stress by absorbing and reflecting solar irradiance, helping to reduce the global warming pollutants associated with waste disposal by turning leaves, grass, woody garden offcuts, and dead garden waste into mulch or compost. Additionally, recycling these “wastes” not only reduces methane emissions from landfills, but also improves a garden’s soil and helps it store more carbon. Furthermore, even if home agriculture is not able to substitute large-scale agricultural productions, its contribution to food production and healthy diets might be enhanced by the organization of small-scale producers within cooperatives that will reach food and grocery retail markets.

### 1.4. Home Gardens’ Structures and Management Impacts on Biodiversity

Biodiversity conservation is usually overlooked in home gardens (and most public gardens also), namely, because other attributes, such as landscape aesthetics, lifestyle, and usefulness, are the gardeners’ (and public managers) objectives [123]. Moreover, home gardens are characterized by being heterogeneous, reflecting their owners’ perceptions and interests, and provide a large variety of small-scale structures that may act as refuge for many species, as well as a valuable network of habitats for meta-populations [124]. On the other hand, the effect of home gardens on local and regional biodiversity patterns relies on the collective action of large numbers of gardeners [125]. Despite the growing awareness of the conservation potential of home gardens, information on “wildlife gardening” and/or “ecological gardening” has been subjected to limited research associated with presumptions of their low ecological value and limited access to researchers [126]. However, ecological and wildlife gardening is characterized by “organic” and/or more sustainable practices and by the creation of habitats for wild species [127]. Some works pinpoint the growing importance of home gardens for the conservation of species and varieties of crops extirpated from the countryside by intensive agricultural and forestry practices [28,128,129]. Additionally, the main features interlaced with animal diversity in gardens are plant-species richness, vegetation structure, plant-species origin, and type of management [130], but more effort should be made for clarifying what really holds true [131].

Home gardens include different “spaces” that fulfil people’s needs and beliefs, which might be considered different “habitats” [132]. Additionally, and when compared with actual intensive and monocultural agricultural landscapes, a high heterogeneity in a rather small area can be observed [133]. This creates opportunities for several species that, along its life cycle, require different resources and habitats [134]. In this way, we have disentangled the several archetypal spaces when discussing the links between home gardens and biodiversity, namely, the vegetable garden; the flower garden; the lawn, trees, shrubs, and hedges; the pond; and paved and constructed areas (Figure 3). The vegetable garden (Figure 3a) is an area usually separated from the rest of the home garden, a source of herbs, vegetables, and fruits; it is also often a structured garden space with a design based on a repetitive geometric pattern, usually incorporating permanent perennials or woody-shrub plantings and annuals [135]. In terms of crop biodiversity, vegetable home gardens are especially diverse, integrating several plant species and, in many cases, non-commercial and local vegetable varieties [28]. Additionally, as in most cases the production is for self-consumption (with exceptions), the use of pesticides, herbicides, and fertilizers is reduced when compared with intensive, commercial productions [136]. The various operations (ploughing, weeding) reduce the number of wild species able to use these areas, and most are generalist species (often plagues) that take advantage of specific plant’s abundance and the lack of predators [137]. However, some species facing serious decline are particularly dependent on vegetable gardens [138]. The flower garden (Figure 3b) is an area where flowers are grown and displayed for their colors and scents. Annual, biennial, and perennial flowers, traditionally associated with native medicinal and condimentary plants, are expanded, at present, to incorporate many others selected by taking into consideration a sequence of bloom and consistent color combinations through varying seasons [139]. At present, great plant “biodiversity”, linked with several thousands of species and varieties, can be found in flower gardens, most with an exotic provenance [140]. Additionally, the pressure for beauty has produced larger and more colorful flowers whose attractiveness to pollinators and many other organisms is, in general, far less their wild ancestors [141]. As an example, most modern rose variety (*Rosa* sp.) selections are related to the conversion of stamens into “petals”, but also by expanding the flowering season through the hybridization and selection of species from several origins at the expense of pollen production and functional nectaries [142]. Nevertheless, flower-bed structures, particularly when associated with native “wildflower” annuals and perennials, could be an excellent contribution to halt the decline in wildflowers and pollinators (and insects in general and many other invertebrates and small vertebrates), but also attract auxiliary organisms that feed upon garden pests [141,143,144]. Most gardens include lawns (Figure 3c) dominated by grasses (monocots), subject to weed and pest control, maintained in a green stage (e.g., by watering), and regularly mowed to ensure an acceptable length for aesthetic and recreational purposes [145]. Even if they might appear a dull monoculture, most include several species of grasses, adapted to diverse environmental conditions and periods of the year. Additionally, many other plants (weeds) adapted to the periodic mowing grow, and a diversity of detritivores, such as springtails (Collembola) that attract predators, such as spiders and ground beetles (Carabidae), become particularly abundant [146]. Anyway, mown laws are very poor in terms of biodiversity, and their intensive management might pose significant risks to several invertebrates and vertebrates (e.g., arthropods’ mortality, vertebrates’ poisoning, and even vertebrates’ mortality) [147,148]. If left uncut for longer periods, the lawn rapidly turns into a (kind of) meadow, which is an incredible hotspot of biodiversity by attracting several species of invertebrates, birds, small mammals, and fungi (e.g., mushrooms) [149,150]. Meadows are a fast-declining habitat in the countryside that could be partially compensated by home garden “wild” lawns [131]. Nevertheless, in regions with water shortages, alternatives to grass lawns using cover plants (e.g., *Hypericum* sp., *Hedera* sp.) might contribute to decrease the impact on water resources and contribute to the conservation of wetlands in the surrounding areas [151,152,153]. Another option to reduce water and agrochemical use is the replacement of grass lawns with artificial lawns constructed from synthetic polymers (plastics), but with significant impacts on home garden biodiversity [154]. Trees, shrubs, and hedges (Figure 3d), both clipped and unclipped, are often used as ornaments in the layout of gardens to enhance a garden’s privacy (e.g., buffer to visual pollution), to create shade/windbreaks for modulating microclimatic conditions, and for producing diverse types of fruits [155]. Woody species, both deciduous and evergreen, are also recognized for their great value to the landscape and wildlife, mostly when the plants are native, older, and are less clipped, namely, by the enhancement of refuge spots (e.g., nesting locations), flower resources, and fruit production [128,156]. These are also among the best locations for the creation of micro-structures and micro-habitats for wild species, such as bird tables, bird and bat boxes, amphibian refuge spots, arthropod boxes, small mammal houses, and even dead hood piles [126,157]. Nevertheless, all woody species play a considerable role in providing shelter for fungi, shade plants, smaller animals, such as birds and mammals (including bats), and insects [158]. Their upscaling, when considering the net of woody species of different home gardens and other trees in the landscape, creates a network of green corridors for many uncommon species in urban and rural areas [126,159]. Additionally, ecosystem services include reducing soil loss and pollution, the regulation of water supplies, and organic carbon storage, critical to the environmental homeostasis of landscapes [160,161]. A garden pond (Figure 3e) is a water feature constructed in a garden or designed landscape, normally for aesthetic purposes, to provide a wildlife habitat, for fish production, or for swimming. The pond is considered the biodiversity “hotspot” of a home garden: nothing beats it in attracting the widest range of species [162]. The diversity of resident amphibians, insects (e.g., dragonflies and water beetles), mollusks, plants, and the usefulness to birds, bats, and small mammals is unquestionable [163]. Considering that amphibians are the vertebrates with the highest rate of extinction, garden ponds might actually be considered relevant for metapopulation conservation [163]. The size of the pond, but specially the non-occurrence of “aquarium” exotic fish (and sometimes invasive plants and turtles), might make the difference in the biodiversity present: fish are particularly aggressive by attacking most organisms and dysregulating food webs, and garden ponds can be pathways for the spread of invasive, non-native plants [164,165]. Ponds are also particularly relevant for environmental education, considering that several organisms are easily spotted along its complex life cycles (e.g., metamorphosis) [166]. Additionally, permanent and temporary ponds are extremely threatened in the countryside, namely, by the compound effects of agricultural intensification in the most productive areas (increase in water consumption and the depletion of water resources) or by agricultural abandonment in less productive ones (e.g., the lack of maintenance of traditional water reservoirs) [167]. Paved and constructed areas (Figure 3f) are impervious surfaces dominated by concrete, asphalt, brick, tile, bitumen, timber, or similar materials, encompassing the walls, courtyard, decking, footpath, driveway, or street access surrounding a house. Even if they seem to be a desert, by comparison with the other “green or blue” infrastructures previously discussed, they possess unique organisms and derive several advantages for many others [168]. Several mosses, hepatics, ferns, and some rock plants are especially diverse in this habitat [169]. Additionally, unique species of arthropods and mollusks are located here, while wild bees, bats, and birds (namely, Hirundinidae—swallows and martins—and Apodidae—swifts) might use them as breeding places [170]. Especially important are old stone walls, with their holes and crevices that mimic stone areas that might be relevant habitats for a diversity of species, including mammals, reptiles, and amphibians [171]. New techniques, such as green roofs and walls, have recently emerged as promising conservation tools, and they offer promising additional opportunities to several species [172].

### 1.5. Home Gardens’ Contribution to Sustainable Development Goals

The authors defined home gardens as gardens that might be characterized by their location, near or around a family house, for their (mostly) private use with a scope linked with the families’ conceptions and needs. We have considered that their contribution to health and wellbeing can be separated within two major influences, nature exposure and outdoor’ stimulation, renowned for their positive physiological and psychological benefits and the physical exercise associated with gardening, providing strength improvement, calorie burning, and, in general, better physical and mental health (Table 3). Concerning diets, their role in boosting food diversity and nutrition should not be disregarded. We highlighted that for low-income and/or isolated regions, this is particularly relevant, including traditional medicine production (Table 3). The authors considered that the possibility of home gardens acting as carbon sinks depended on several factors linked with management; more research is needed to understand this potential function (Table 3). Biodiversity conservation is a complex issue, but with careful management, design, correct size, and habitat creations, the home gardens matrix might contribute to sustain “wild” species metapopulations, by the planting of native species (e.g., tree species) or by the resources associated with the “habitats” created by the gardener (Table 3).

## 2. Complementary Remarks on the Risks and Drawbacks of Home Gardens and Home Gardening in the Scope of Their Contribution to Sustainable Development Goals

### 2.1. Risks to Health and Wellbeing of Home Gardens and Home Gardening

Musculoskeletal injuries are very common in individuals who farm the land [44], although addressing this type of injury in gardening is still very limited in the literature [48]. Some authors indicated that the causes for their manifestation are related to the presence of inadequate work practices (repetitive and performed for a prolonged period), biomechanical factors (improper handling of gardening tools, lifting and carrying heavy loads, repetitive flexion movements of the spine, and excessive movement of the lumbar region or neck), use of poorly ergonomic tools, fatigue, and poor physical fitness [173,174,175]. Low-back pain is very common and is aggravated by age, crop type, stress levels, and the presence of previous occupational injuries [175]. The manual and repetitive activities associated with gardening, such as planting, spraying, sweeping, and using shears, can also lead to wrist and hand injuries. Short-rest breaks, the use of ergonomic tools, and elevated flowerbeds are some of the strategies that can help reduce the symptoms of fatigue and musculoskeletal discomfort associated with gardening. Other health risks associated with gardening involve exposure to chemical substances through skin contact, ingestion, or inhalation with dermatological, gastrointestinal, neurological, oncological, respiratory, and endocrine effects [176]. Elderly gardeners and immunosuppressed individuals are particularly sensitive to Legionellosis, an infectious disease caused by exposure to Legionella bacteria present in compost submitted to high temperatures [177]. Handling it might cause the release of microorganisms and bioaerosols, providing ideal conditions for the growth of fungi that lead to non-allergic, immuno-allergic (rhinitis, allergic asthma), and inflammatory reactions [178]. Cuts and wounds resulting from handling thorny plants and power tools and gardening equipment facilitate the entry into the body of spores of *Clostridium tetani* bacteria, resulting in the onset of muscle spasms, cramps, and even convulsions. Some plants and insects (bees, wasps, and red ants) can also cause allergic reactions in some individuals. Tick bites, very common in gardens [179], affects the joints and nervous system [180]. Moistening dry compost before turning or using it, wearing gloves, and keeping your arms covered when pruning plants likely to cause irritation may minimize the occurrence of some of these health risks. However, considering the state of the art to date, further research is needed on this issue, namely, by measuring exposure, understanding the underlying mechanisms, and demonstrating causality [181]. This is even more incomplete when it comes to home gardens and home gardening: research is needed to understand the real benefits of the spaces created and management practices on human health and wellbeing, something that ought to be performed by integrated teams linking ecologists and health and social scientists with gardeners. With strong (and hopefully) positive results, policies might be developed to promote home gardens and home gardening (and gardens and gardening in general) in the scope of the one-health approach [181]. In fact, home gardens might contribute to halt habitat degradation/destruction and biodiversity loss, mitigate locally ongoing climate change, and contribute to several human wellbeing and health benefits of experiencing nature.

### 2.2. Food Provision, and Nutritional and Carbon Footprint Risks

The positive effects of home gardens and home gardening on food provision, diet diversity, nutrient supply, and carbon footprint were highlighted in the previous section. For example, home gardens were presented as relevant to obtain a continuous supply of daily foods for households in remote locations [36], supplementing diets with proteins, vitamins, and minerals, and thus contributing to food security, food diversity, and nutrition. However, drawbacks were reported: gardens are often located near roads or intensive agricultural areas, which are more susceptible to be contaminated by heavy metals [182,183,184] and organic pollutants (for example: polycyclic aromatic hydrocarbons (PAHs)), antibiotics, and petroleum products and pesticides [183]. In this situation, gardeners may be exposed to these substances, which are an important set of constraints highlighted by several research studies [185]. In fact, growing foods within or near the main roads, factories, or intensive agricultural fields increases the chance of high concentrations in the soil of potentially toxic elements, such as As, Cr, Cu, Fe, Mn, Ni, Pb, Sn, and Zn, which can be obstacles to produce safe and healthy fresh products [186,187,188]. Nonetheless, several solutions can be implemented to minimize the potential risk of soil contamination, such as building raised beds for the crops, using amendments to stabilize contaminants in soil, adding thick layers of organic matter to the soil (i.e., providing a physical barrier to contamination), replacing contaminated soil with clean soil, or even using plant species that extract, degrade, contain, or immobilize the contaminants in soil [189]. Thus, despite the potential risks of contaminants in soils, several practices are available at present to (partially) overcome this problem. Moreover, whenever pesticides are used (insecticides, fungicides, and herbicides)—gardeners often lack the training for how to use them safely—harmful effects to human health could increase. Even if several “problematic” pesticides are banned from developed countries, in many others, they would still be commonly use. Moreover, a recent research revealed that in the UK, growers can easily purchase unauthorized pesticides online, including atrazine, a herbicide which has been banned for sale in the EU for more than a decade [190]. Another important criticism of home gardens and home gardening is linked with their food-provision role: several authors suggested that only small and modest contributions to overall food and nutritional needs are fulfilled [191]. Moreover, the majority of studies involving home gardening only address the potential of urban soil for food production and how much land or what types of soils would be required to feed the city’s population. In fact, comparisons are sometimes difficult to establish, since methods of crop production and types of crops differ among studies. The lack of data also complicates the comparisons between potential urban and rural home gardens and home gardening, but a question always raised is about the amount of food supplied by gardens to households. From our point of view, the objectives of domestic and home gardens should not be to provide the complete needs of nutrients to households, but to complement them, since extensive areas of farming already exist for this purpose. On the other hand, critics always refute the idea that gardens can effectively contribute to reducing the carbon footprint, suggesting that garden species need extra care with fertilization, watering, and sanitary treatments [191]. The excessive use of mineral fertilizers, especially nitrogen and potassium, might end up in groundwaters, but also the accumulation of pesticide residues in soils and foods, and groundwater depletion, are among the other issues and problems raised [191]. Nonetheless, the majority of criticism and drawbacks reported, even opportune as reflection points, can be minimized using correct “farming” practices. The use of organic fertilizers, green and organic amendments, natural substances, or natural products for crop sanitary treatments, the rational use of irrigation water (only when necessary) have already been proven to reduce the negative impacts on the environment [118]. The widespread use of mulching or compost, ground cover, vegetables and fruits in raised beds (filled with an uncontaminated soil), no tillage, and sowing annual plants away from busy roads are practices to be considered [117]. All these practices can also contribute to the recovery of degraded soils in gardens, but also capture different forms of atmospheric carbon, contributing in this way as a carbon sink [116].

### 2.3. Biodiversity and Nature Conservation: The Downside of Home Gardens

Most gardens are not suited for the conservation of species with special requirements of area, soil, climate, or habitat [192]. In fact, urban sprawl is one of the most threatening factors, by reducing natural habitats area and their ecological status, i.e., natural-habitat conservation should be the priority [25]. Nevertheless, in the advent of an increasing urbanized and agriculture-intensive world, wildlife gardening could create, within a small area, a diversity of microhabitats suited for several species [192]. Conversely, several of the species selected by gardeners or attracted to live in our home gardens (e.g., cats, naturalized and/or invasive species) may pose huge threats to our wild neighbors by spreading infectious diseases, predating several vertebrates and invertebrates, but also competing for space, nutrients, and light [193,194,195]. In fact, a relevant drawback related to home gardens and home gardening is linked to the chosen species, namely, the potential of exotic species becoming invasive [196]. In fact, several plants and animals (and their associated parasites) brought to the garden in order to increase its beauty (e.g., colorful flowers) escape and become invasive in the wild habitats, creating considerable challenges for conservationists by competing with native species, changing web links and fire regimes, and spreading new diseases [144]. Additionally, downfalls created by our longing for beauty, “cleanness”, and pest control within our home garden havens, by over-adding fertilizers and pesticides, might create traps and mortality events for several non-target species [137,197,198]. A gardener’s education and garden-center consultations should be a priority to tackle this problem [199]. Another relevant aspect, namely, in regions with water scarcity (e.g., Mediterranean region), is the preconception of “green” gardens (e.g., lawn) that need high quantities of water (and chemicals) to maintain their features [200]. Apart from the costs associated with water consumption, this water is, many times, deviated from subterranean waters, wetlands, or associated with the construction of reservoirs that impact natural habitats further [201]. Education could again pave the way for more sustainable gardens, by including novel irrigation techniques using gray waters, xeriscape concepts, and mimicking regional natural habitats by choosing native species adapted to the local climate and soils [196].

### 2.4. Risks and Drawbacks of Home Gardens and Home Gardening to the Implementation of Sustainable Development Goals

Recognizing the potential interest of home gardens and home gardening for health and wellbeing, namely, for elderly people, the authors also would like to stress that musculoskeletal injuries are a common problem that could be reduced by specific gardening education directed to postural techniques and tool use (Table 4). On the other hand, the correct handling of chemicals (e.g., limiting the access to accredited gardeners) might prevent the occurrence of toxicological effects, while proper clothing and hygiene can also minimize skin lesions from plants and arthropods and reduce infection by microorganisms. The authors recognized the risks of contamination of home gardens located near urban/industrial or intensive agriculture areas with metals and/or pesticides (but also in some cases by the excessive use of agrochemicals by the gardener) that might end up in the legumes, fruits, and groundwater (Table 4). Several techniques are available, ranging from mulching to organic farming and cover crops, to help in tackling the previous problems, but further studies are needed to prove efficacy. In our opinion, these techniques might also reduce water consumption and contribute to capturing carbon from the atmosphere. The biodiversity of home gardens might be enhanced by choosing the right species and correct management techniques, and is by no means comparable to the biodiversity found in natural habitats (Table 4). Additionally, home gardens might significantly impact natural ecosystems’ functioning and biodiversity: the introduction of alien, invasive species and water consumption in arid and semi-arid regions, but again, the environmental education of gardeners might make a difference (Table 4).

## 3. Discussion

Even if the maintenance of a small home garden is mostly associated with low- and moderate-intensity activities, it can serve as a gateway within a plexus of outdoor activities, contributing to reduce the seasonality of physical-activity levels that usually tend to occur under good weather conditions [202]. Moreover, for the elderly population with reduced mobility, significant improvements to their health condition and mental wellbeing were noticed with gardening practices [203]. In this way, the backyard becomes a potential outdoors gymnasium, upscaling physical activity and nature integration for individuals that face barriers to the practice of physical activity, but whose participation is a priority [203]. Since different management activities are performed during specific seasons and linked with diverse garden structures and species, gardening encourages physical activity throughout the year [51] by motivating the adoption of recommended levels for healthy lifestyles, viz., 150 min of moderate-intensity physical activity per week [47,51,185,203]. This recommendation can produce considerably positive effects in reducing the risk of several diseases, such as obesity, hypertension, type 2 diabetes, cardiovascular disease, and even some types of cancer [204].

In fact, growing ornamental plants for aesthetic purposes and/or fruits/vegetables for home consumption is one unpretentious way of interacting with flora and fauna, while promoting a greater interest and knowledge of nature-related issues [24]. Green spaces in balconies, terraces, backyards, or other areas increase an individual’s exposure to natural elements and biota while promoting physical activity, regardless of socioeconomic status [205]. Home gardens and gardening also have the potential for changing behaviors, including the preference for healthy diets and de-tress activities, thus contributing to the prevention and control of chronic diseases [206]. Additionally, gardens are able to reduce air, noise, and thermal pollution, while providing important ecosystem services, such as oxygen production and water percolation, in urban areas [207].

Nonetheless, it is still not clear what the mains aspects of home gardens and home gardening are that promote human welfare. In recent decades, the efforts of researchers have been dedicated to explore the attitudes of communities to domestic gardening, and how gardening is seen as a health-intervention strategy. Home gardening and small gardens are a complex multi-factorial activity, having direct and indirect impacts on the health and wellbeing of those taking part in it [208]. Moreover, consumers see the garden (and gardening) as a way to preserve plants and green spaces, as well as an activity that has considerable contributions to wellbeing [209]. The authors reported that consumers perceived gardens and gardening as spaces and an activity to relax, to find restoration from daily stress, engaging in physical activities with spiritual meaning [210,211]. The improvement to wellbeing in older people was also pointed out as a benefit of home gardens and home gardening [44,212].

Even if home gardens are often overlooked within biodiversity conservation, their cumulative impact should not be underestimated [28]. Being artificial, gardens encompass mostly generalist and adaptable habitats and species that used to be considered as “uninteresting” by researchers and, in this way, were understudied [153]. The trends in the last decade have shown that several of these generalist habitats and species are rapidly retreating and listed as habitats and/or species of conservation concern [213]. In fact, for many species associated with agroecosystems that are more and more intensified and monocultural, gardens could work as conservation islands [214]. This is discussed in several forum sites and wildlife-gardening publications, namely, highlighting that gardens should be considered in the mainstream of conservation thinking [215].

To finish our viewpoint, we suggest looking at home gardens by considering a concept similar to high-nature-value farming (e.g., [216]), which recognizes the importance and special status of traditional agricultural systems and practices for nature conservation within the rural landscape at present. Furthermore, we would like to extend the “nature” focus of the previous concept by also including food provision and the active use of the outdoor environment, translated into social (reduced isolation, improved social networks), mental (reduced stress and depression, improved cognitive function), physical (increased physical activity and weight control), nutritional (quality and diversity of food items), ecological (conservation of habitats and species), and carbon footprint (sink habitats) improvements within landscapes. In fact, home gardens provide opportunities for leisure and self-expression, encourage creativity, skill development, and the adoption of pro-environmental behaviors, all with valuable contributions to increasingly homogeneous ecosystems (both rural and urban) and their inhabitants [24]. Through home gardening, urban and rural populations could develop extra proficiency related to plants and nature in general, which could increase environmental consciousness through their involvement. Public authorities could produce rules and policies to stimulate the contribution of residential home gardens to citizen’s health through eco-therapy, urban agriculture, pedagogical farms, or green/social programs, gauging their contribution to the SDG’s zero hunger, good health and wellbeing, clean water and sanitation, sustainable cities and communities, responsible consumption and production, and climate action and life on Earth. For accomplishing the SDGs, home-garden-management practices may need to be redesigned and accommodated in order to conduct the required research that will foster the transition to a low carbon, climate resilient, and sustainable use of resources [217]. In this scope, the sustainable management of home gardens could be supported by simple indicators that might enlighten their resilience status due to their ability to bridge production, environment, biodiversity, and the associated ecosystem services (e.g., [218,219,220]). Additionally, and based on the cause–effect relationships being conceived to solve focal environmental problems, socio-ecological models might be used to predict the outcome of alternative scenarios in order to support gardeners and local authorities’ decision making (e.g., [221]). In fact, when properly developed and tested, socio-ecological models might enlighten what drives biodiversity and ecosystem functioning at the home garden scale, including the re-use of agriculture wastes, the storage of carbon in soils, the protection of other soil functions and ecosystem services, as well as the link between soils, food quality, and enhanced shelf-life. Since there are important gaps in our understanding of ecosystem services (provisioning, regulating, and cultural services) valuations, the main challenge for predictive research lies in the key interactions between relevant landscape characteristics, management strategies, and SDGs. A special focus on the economic valuation of the ecosystem services is crucial, not only for the methodological challenges involved (e.g., addressing the value of biodiversity, ecosystem resilience, or cultural heritage), but also because the final outputs can be of major interest for managers and policy-makers. From this perspective, we highlighted the interplay between model-based research and the SDGs’ achievements. This evaluation might be a first step to increase society’s recognition of the multifactorial importance of front and backyard home gardens, but is also a possibility to increase our present and future sustainability practices within an increasingly urbanized and monocultural farming world.

## 4. Conclusions

Our supported opinion aimed to describe and discuss the evidence of the effects of home gardens and home gardening on wellbeing and health, nutrition, carbon footprint reduction, biodiversity and nature conservation, fundamental issues for achieving sustainable development goals (SDGs). The objectives were to understand their benefits (and drawbacks), and provide an opinion reinforced by the literature, in order to guide scientists, managers, and policymakers in envisioning home gardens and home gardening as humble but significant strategies in this scope. The strength of our supported opinion was its approach to understanding the breadth of the authors’ opinions on the effects of selected SDGs.

## Figures and Tables

**Figure 1 ijerph-19-13715-f001:**
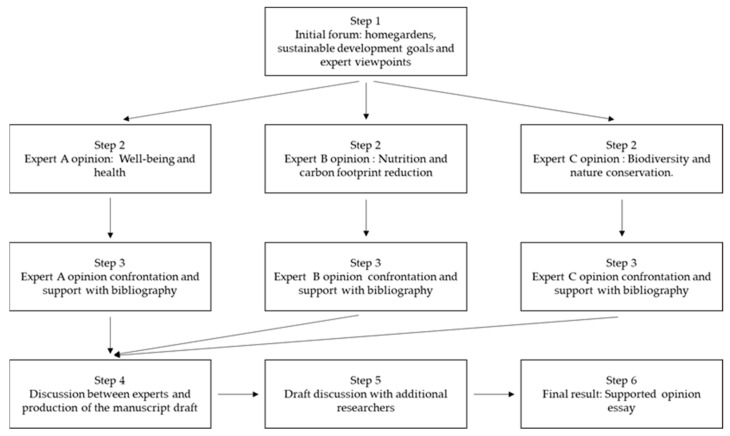
The stepwise framework used to structure our supported opinion essay: step 1, forum between the core researchers, including the definition of home gardens and their significance for sustainable development goals; step 2, core researchers’ drafts concerning home gardens and sustainable development goals; step 3, support and confrontation of core researchers’ ideas and opinions of bibliography; step 4, discussion of different viewpoints and production of the first manuscript draft; step 5, enlarged discussion of additional experts’ opinions; step 6, production of the supported opinion essay.

**Figure 2 ijerph-19-13715-f002:**
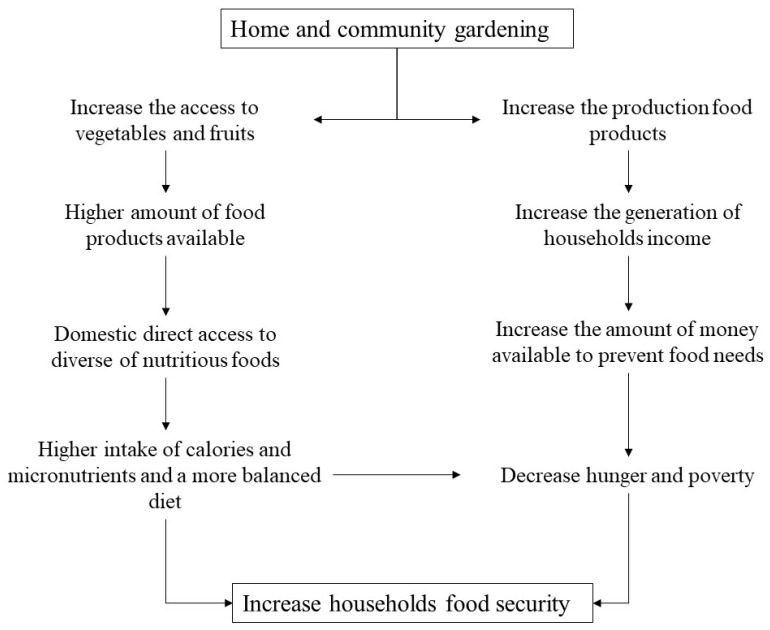
Pathways of contribution of home (and community) gardening to food security.

**Figure 3 ijerph-19-13715-f003:**
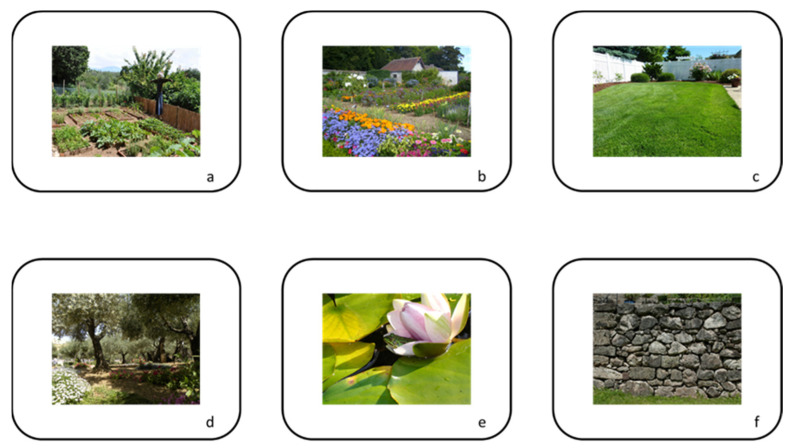
Home garden most usual “spaces”: (**a**) the vegetable garden; (**b**) the flower garden; (**c**) the lawn; (**d**) trees, shrubs, and hedges; (**e**) the garden pond; (**f**) paved and constructed surfaces.

**Table 1 ijerph-19-13715-t001:** Classification of the intensity of various gardening activities. Adapted from [50].

Lawn and Garden	Low Intensity (1.6–2.9 METs)	Moderate Intensity (3.0–5.9 METs)	Vigorous Intensity (≥6 METs)
Digging, spading, filling garden, composting, vigorous effort			7.8
Chopping wood, splitting logs, vigorous effort			6.3
Clearing brush/land, undergrowth, or ground, hauling branches, wheelbarrow chores, vigorous effort			6.3
Laying crushed rock			6.3
Gardening with heavy power tools, tilling a garden, chainsaw		5.8	
Carrying, loading or stacking wood, loading/unloading or carrying lumber		5.5	
Wheelbarrow, pushing garden cart or wheelbarrow		5.5	
Mowing lawn, general		5.5	
Felling trees, small–medium sizes		5.3	
Digging sandbox, shoveling sand		5.0	
Mowing lawn, walk, power mower, moderate or vigorous effort		5.0	
Digging, spading, filling garden, compositing		5.0	
Weeding, cultivating garden, using a hoe, moderate-to-vigorous effort		5.0	
Hopping wood, splitting logs, moderate effort		4.5	
Planting trees		4.5	
Picking fruit off trees, gleaning fruits, picking fruits/vegetables, climbing ladder to pick fruit, vigorous effort		4.5	
Planting seedlings, shrub, stooping, moderate effort		4.3	
Trimming shrubs or trees, manual cutter		4.0	
Raking lawn or leaves, moderate effort		3.8	
Gardening, general, moderate effort		3.8	
Clearing light brush, thinning garden, moderate effort		3.5	
Digging, spading, filling garden, composting, light-to-moderate effort		3.5	
Trimming shrubs or trees, power cutter, using leaf blower, edge, moderate effort		3.5	
Picking fruit off trees, picking fruits/vegetables, moderate effort		3.5	
Weeding, cultivating garden, light-to-moderate effort		3.5	
Carrying, loading or stacking wood, loading/unloading or carrying lumber, light-to-moderate effort		3.3	
Walking, applying fertilizer or seeding a lawn, push applicator		3.0	
Walking, gathering gardening tools		3.0	
Driving tractor	2.8		
Gardening, using containers, older adults >60 years	2.8		
Planting, potting, transplanting seedlings or plants, light effort	2.0		
Watering lawn or garden, standing or walking	1.5		

**Table 2 ijerph-19-13715-t002:** Benefits, pursuits, and encouragement of home gardening related to human diet.

Benefits
Increase in consumption of fruits and vegetables for home gardeners and families
Contribute to food security and potentially enhance livelihoods
Households diet is not totally dependent on the availability of markets
Supplying yearly fresh and nutritious foods
Essential nutrient supplements (minerals, vitamins, and micronutrients)
Young children’s habituation to diverse (and seasonal) diets
Local spices and medicinal herb provisions
Extra income for poor communities

**Table 3 ijerph-19-13715-t003:** Contribution of home gardening to sustainable development goals.

Benefits
Nature-exposure benefits to wellbeing
Outdoor stimulation and exercise benefits to health
Diet diversification, traditional medicine production, and extra income
Carbon sequestration and biodiversity conservation contingent upon management

**Table 4 ijerph-19-13715-t004:** Risks and drawbacks of home gardens and home gardening.

Risks/Drawbacks
Musculoskeletal injuries
Ecotoxicological effects associated with agrochemicals
Infection by microorganisms of scars and skin lesions
Contamination by metals and other toxic substances
Invasive species introductions
Water consumption in arid regions

## Data Availability

Not applicable.

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
