# Peer review of "Contribution of Home Gardens to Sustainable Development: Perspectives from A Supported Opinion Essay"

_ijerph, 2022, doi:10.3390/ijerph192013715_

Round 1

Reviewer 1 Report (Previous Reviewer 2)

The quality of the article in this version is better than in the first version.

Recommendations for authors:  

-it is recommended to write a summary and the opinion of the authors after each chapter.

-it is recommended to prepare new conclusions, make them more specific, and shorten them. Do not present the opinions of other authors in the conclusions, only the results of your own research.

Author Response

Reviewer 1

1 - The quality of the article in this version is better than in the first version.

R1: Dear reviewer, we would like to acknowledge for the important suggestions made by you, that, in our humble opinion, contributed to the increase in the manuscript quality. Thanks also for your comment concerning the manuscript quality.

- Recommendations for authors:

-it is recommended to write a summary and the opinion of the authors after each chapter.

R2: We acknowledge the reviewer suggestions that was considered, by all co-authors, as pertinent and fundamental in an opinion manuscript. In this way, in the end of each main chapter, a summary and the opinion of the authors was added (along with table to resume information):

1.5. Home Gardens Contribution to the Sustainable Development Goals

The authors define home gardens as gardens that might be characterized by their location, near or around a family house, for their (mostly) private use with a scope linked with the families’ conceptions and needs. We consider that their contribution to health and well-being can be separated within two major influences: nature exposure and out-door’ stimulation, renowned for its positive physiological and psychological benefits and the physical exercise associated with gardening, providing strength improvement, calo-ries burning and, in general, a better physical and mental health (Table 3). Concerning di-ets, their role in boosting food diversity and nutrition should not be disregarded. We high-light that for low income and/or isolated regions, this is particularly relevant, including traditional medicine production (Table 3). The authors consider that the possibility of home gardens acting as carbon sink depends on several factors linked with management; more research is needed to enlighten this potential function (Table 3). Biodiversity conservation is a complex issue but, with careful management, design, correct size and habitats creation, home gardens matrix might contribute to sustain “wild” species metapopulations, by the plantation of native species (e.g. tree species) or by the resources associated with the “habitats” created by the gardener (Table 3).

Table 3. - Contribution of home gardening to the Sustainable Development Goals

Benefits

  • Nature exposure benefits to well-being
  • Outdoors stimulation and outdoors’ exercise benefits to health
  • Diet diversification, traditional medicines production and extra income
  • Carbon sequestration and biodiversity conservation contingent upon management

and

2.4. Risks and Drawbacks of Home Gardens and Home Gardening to the implementation Sustainable Development Goals

Recognizing the potential interest of home gardens and home gardening for health and well-being, namely for elderly people, the authors also would like to stress that musculoskeletal injuries are a common problem, that could be reduced by specific gardening education directed to postural techniques and tools use, among other (Table 4). On the other hand, correct handling of chemicals (e.g. limiting the access to accredited gardeners) might prevent toxicological effects occurrence while proper clothing and hygiene can also minimize skin lesions from plants and arthropods and reduce infection by microorganisms. The authors recognize the risks of contamination of home gardens located near ur-ban/industrial or intensive agriculture areas with metals and/or pesticides (but also in some cases by the excessive use of agrochemicals by the gardener) that might end the legumes, fruits and groundwater (Table 4). Several techniques are available, from mulching to organic farming and cover crops, to help tackling the previous problematics, but more studies are needed to prove efficacy. In our opinion, these techniques might also reduce water consumption and contribute to capture carbon from the atmosphere. Biodiversity of home gardens might be enhanced by choosing the right species and correct management techniques, is by no means comparable to the biodiversity found on natural habitats (Table 4). Additionally, home gardens might impact significantly natural ecosystems functioning and biodiversity: the introduction of alien invasive species and water consumption in arid and semi-arid regions but again, environmental education of gardeners might make a difference (Table 4).

Table 4. – Risks and Drawbacks of Home Gardens and Home Gardening

Risks/Drawbacks

  • Musculoskeletal injuries
  • Ecotoxicological effects associated with agrochemicals
  • Infection by microorganisms of scars and skin lesions
  • Contamination by metals and other toxic substances
  • Invasive species introductions
  • Water consumption in arid regions”

-it is recommended to prepare new conclusions, make them more specific, and shorten them.

R3: We have considered this suggestion particularly important, namely for a more assertive take-home message. In this way, a new short conclusion was introduced (and a reorganization of the discussion was made). The new conclusion:

“4. Conclusion

Our supported opinion aimed to describe and discuss the evidence of the effects of home gardens and home gardening on the well-being and health, nutrition, carbon foot-print reduction, biodiversity and nature conservation, fundamental issues for achieving the sustainable development goals (SDGs). The objectives were to understand its benefits (and drawbacks), provide an opinion reinforced by the literature, in order to guide scientists, managers and policy-makers in envisioning home gardens and home gardening as a humble but significant strategy in this scope. The strength of our supported opinion was its approach to understand the breadth of the authors’ opinion on the effects on selected SDGs.”

 - Do not present the opinions of other authors in the conclusions, only the results of your own research

R4: Again, many thanks for your suggestion. In the new conclusion of our opinion, we did not consider other authors’ conclusions.

Reviewer 2 Report (New Reviewer)

The information in the manuscript does not meet expectations or what is expected to be read when reading the title "Neglected socio-ecological conflict points: contribution of family gardens to sustainable development" that is, the information or structure of the information that makes up the manuscript is not adequate to know disregarded Social-Ecological Hotspots: Contribution of Home Gardens to Sustainable Development. Check which term is correct: Home Gardens or family gardens. In certain paragraphs there is talk of family gardens in others of gardens. The paragraph that begins in L105 where the comparison and definition of a common vision about family gardens is developed. The information in the paragraph is vague about what the subtitle refers to. 1.2 how did they conduct the assessment of the health and well-being contributions of gardening activities? 1.3. Home gardening and urban agriculture to promote food and nutrition security and reduced carbon footprint. The information in some ways is compared to large productions, however, it is not clear how it is that an orchard can replace large-scale production. It is agreed that every house cannot have an orchard, so you cannot talk about a reduction in emissions in the dimensions that are written in the writing. It would be clearer if they wrote of a percentage of emission reduction per square meter of orchard compared to square meters of large-scale production for example. 1.4. Implication of the structure and management of gardens for biodiversity, places a lot of information in such a way that it is not clearly identified what it intends to describe. 2.1. Health and Well-being, it refers again to sub-item 1.2. talking about the benefits you have with respect to well-being. 2.2 again describes the carbon footprint issue I mention in 1.3. 2.3 the approach of the sub-theme is not clear with respect to what has been written, Biodiversity and Nature Conservation from where or from whom? Of the orchards or of the environment. It reads a bit contradictory information. Final Remarks or conclusions should not be based on other works (References), rather, it must be written based on the results obtained and the objective of the manuscript. It is recommended to review wording.

Author Response

Reviewer 2

The information in the manuscript does not meet expectations or what is expected to be read when reading the title "Neglected socio-ecological conflict points: contribution of family gardens to sustainable development" that is, the information or structure of the information that makes up the manuscript is not adequate to know disregarded Social-Ecological Hotspots: Contribution of Home Gardens to Sustainable Development.

R5: Dear reviewer, many thanks for your comment concerning our title / expectations for our manuscript. Even if the reviewer has considered one of the meanings of the word hotspot, “danger of fighting or conflict”, the authors have used the term within a completely different meaning, that is “exciting place where there is a lot of activity” and/or “any place regarded as a center for a specified activity or interest”

Please see:

https://www.collinsdictionary.com/dictionary/english/hot-spot

Anyway, we have considered the reviewer commentary important, in the sense that the title should be more objective and straightforward to the reader. In order to increase the clarity of our title we propose changing it to a new one, hoping that now is more suited for the manuscript:

Contribution of Home Gardens to Sustainable Development: perspectives from a supported opinion essay”

Check which term is correct: Home Gardens or family gardens. In certain paragraphs there is talk of family gardens in others of gardens.

R6: Dear reviewer, again many thanks for your pertinent comment. The terms are partially superimposed, and partially interchangeable…most home gardens are linked with families and most family gardens are in fact home gardens:

Please see:

https://link.springer.com/chapter/10.1007/978-3-319-67283-0_4

https://agricultureandfoodsecurity.biomedcentral.com/articles/10.1186/2048-7010-2-8

Anyway, to tackle this issue, all manuscript was revised and now is mostly focusing on home gardens. New text along all manuscript highlights our intention to focus on home gardens, and tries to show contact points with other “types” of gardens (family, community, and others) (example shown below is for the definition of home garden, but several small changes were made along the text in this scope):

1.1. Home garden definition and methodological framework

1.1.1. Home garden definition

A garden is a planned space, usually outdoors, set aside for the cultivation, display, and enjoyment of plants and other forms of nature [35]. Within gardens, home gardens are small areas (usually far below 1 hectare) surrounding the residential parcel, usually associated to family use (most home gardens are family gardens), characterized by combinations of various perennial and annual plants, sometimes in association with domestic animals and might include additional infrastructures such as ponds, greenhouses, green roofs, among other [35].

The paragraph that begins in L105 where the comparison and definition of a common vision about family gardens is developed. The information in the paragraph is vague about what the subtitle refers to.

R7: Again, many thanks for your important remarks. We have changed the subtitle, and introduced changes in the paragraph so it becomes more obvious that we are trying to explain the methodological steps made to “built” our opinion manuscript. We will, of course, introduce more changes if the reviewer considers that they are still incomplete. The new paragraph:

1.1.2. Supported opinion methodological framework

A pre-opinion online and face to face forum took place during the 2th semester of 2021, involving three co-authors of the manuscript, in order to debate the personal views and importance of home gardens to tackle several of the sustainable development goals. For the purposes of the opinion manuscript, as each one of the co-authors’ expertise is focused on different scientific domains, namely healthy lifestyles and green exercise (Helena Moreira), food production and food quality (Alfredo Aires), agroecosystems, biodiversity and conservation ecology (Mário Santos), the idea of the preliminary discussions was to compare and define a common view concerning home gardens and to guide the workflow associated [39]. An expert draft was produced with his/her viewpoint, sharing and con-fronting with the other co-authors and justifying, whenever possible, their expert opinion using relevant bibliographic information. Afterwards, during the 1th semester of 2022, the forum was enlarged to include additional researchers’ opinions (João Alexandre Cabral, Ronaldo Gabriel, Andreia Teixeira and Rita Bastos), comments and discussion of ideas, reaching consensus in several issues [40]. This former step enabled introducing complementary remarks and risks, that were integrated in the discussion. The final manuscript highlights the key themes and perceptions that emerged during the discussions, based on sound evidence and research, usually termed a supported opinion essay (Figure 1).”

1.2 how did they conduct the assessment of the health and well-being contributions of gardening activities?

R7: Dear reviewer, as a supported opinion manuscript, the expert author responsible by each subtheme first introduced their opinion, while searching links with publications to support its concepts and ideas:

1.1.2. Supported opinion methodological framework

"….An expert draft was produced with his/her viewpoint, sharing and confronting with the other co-authors and justifying, whenever possible, their expert opinion using relevant bibliographic information.

and

“1.2. An Appraisal of Health and Well-being Contributions from Home Gardens and Home Gardening Activities

Several of the issues below are linked to home gardens and gardening, even if they should also be considered with other types of “nature immersion” and outdoor activities.”

1.3. Home gardening and urban agriculture to promote food and nutrition security and reduced carbon footprint. The information in some ways is compared to large productions, however, it is not clear how it is that an orchard can replace large-scale production. It is agreed that every house cannot have an orchard, so you cannot talk about a reduction in emissions in the dimensions that are written in the writing. It would be clearer if they wrote of a percentage of emission reduction per square meter of orchard compared to square meters of large-scale production for example.

R8: The authors agree with the pertinent reviewer suggestion. In fact, in the revised manuscript we have consider home gardens crop production mostly supplementary, without disregarding that, in specific locations and contexts, home gardens might be particularly important, outweighing commercial crops. Concerning percentage of emission reduction, it was considered by the authors that so many variables might influence the comparison between large-scale (type of crop, distance from the market, type of crop production, among others) and home gardens (type and management, among others), that it would be not accurate to produce a reduction percentage table. Also, in the revised manuscript we have disentangled the benefits from the drawbacks (or dubious questions) concerning food production and emission reduction. Some examples below:

1.3. Home Gardening (and Urban Agriculture) contribution for Dietary Diversity and Carbon Reduction Footprint

…Different studies have associated home gardening with a wide range of ecosystem services such as supplying small markets with high quality fruits and vegetables, and employment opportunities [90,91]. In fact, divergent economies (countries from North America, South America, Europe, Australia and Asia) envision home gardens (domestic agriculture) contributing to a reduction in the world food crisis [92]. Consumers often perceived home agriculture as a supplementary strategy to assure food security, since it can be a source of income while providing direct access to a larger number of nutritionally rich foods (vegetables and fruits) [93]. Increase stability of household diets against seasonality or other temporary shortages was also pointed out [94,95,96].

…..

In the literature, it is possible to find several studies in which home and community gardens are considered inefficient, costly, and without any offer regarding a complete solution to food insecurity [114]. We can however look at the problem from another perspective: instead of considering home and community gardens as a solution to address food insecurity, we may consider them as a part of a broad answer to address a much larger issue of offering a diversity of nutritious foods, as well as the opportunity for positive health outcomes.

….

In addition, greenhouse gases (GHG) footprint of consumers through the conventional agribusiness systems is far higher than of home productions [125]. Conversely, other re-search suggests that the impact of home gardens on the environment may depend on the specific management and cultivation methods used [126], which in turn might have negative consequences in terms of the overuse of fertilizers and/or the GHG emissions. Home composting from gardening wastes can also produce methane and nitrous oxide, which are strong GHG [127-129]. Nonetheless, GHG reductions were observed in gardening communities, compared with conventional systems, particularly when vegetable production replaces lawns [124]. In developed countries, post-harvest processes such as storing, refrigeration and transportation over long distances have high GHG emissions, comparable to the production processes [130]. These postproduction emissions are strongly reduced when vegetables are grown near the places where they are consumed, as in the case of home gardens.”

And

2.2. Food Provision, Nutritional and Carbon Footprint Risks

Moreover, the majority of studies involving home gardening only address the potential of urban soil for food production and how much land or what types of soils would be re-quired to feed the city’s population. In fact, comparisons are sometimes difficult to estab-lish, since methods of crop production and types of crops differ among studies. The lack of data also complicates the comparisons between potential urban and rural home gardens and home gardening, but a question always raised is about the amount of food supplied by gardens to the households. In our point of view, the objectives of domestic and home gardens should not be to provide the complete needs of nutrients to households, but to complement, since extensive areas of farming already exist for this purpose. On the other hand, critics always refute the idea that gardens can effectively contribute for reducing the carbon footprint, mentioning that garden species need an extra-care with fertilization, watering and sanitary treatments [211].”

1.4. Implication of the structure and management of gardens for biodiversity, places a lot of information in such a way that it is not clearly identified what it intends to describe.

R9: Again, many thanks for your question and doubts. The idea of this section was to highlight the general importance of home gardens for biodiversity. Considering that a home garden might be divided within different types of “habitats”, our idea was to explain some of the most common ones and its associated biodiversity. We have tried, by changing the title and part of the text, that this is more explicit to the reader. We have also changed part of the following sections (2.3) to discuss the problems for biodiversity associated with home gardens and home gardening management. We are, of course, willing to add more information if the reviewer finds it incomplete.

1.4. Home Gardens’ Structure and Management Impacts on Biodiversity

Biodiversity conservation is usually overlooked in home gardens (and most public gardens also), namely because other attributes such as landscape aesthetics, lifestyles and usefulness are the gardeners’ (and public managers) objectives [131]. Anyway, home gar-dens are characterized by being heterogeneous, reflecting owners’ perceptions and interests, and provide a large variety of small-scale structures that may act as refuges for many species, as well as valuable network of habitats for meta-populations [132]. On the other hand, the effect of home gardens in local and regional biodiversity patterns relies on the collective action of large numbers of gardeners [133]. Despite the growing awareness of the conservation potential of home gardens, information on “wildlife gardening” and/or “ecological gardening” has been subjected to limited research, associated with presumptions of their low ecological value and limited access to researchers [134]. ….”

and

2.3. Biodiversity and Nature Conservation: The Dark Side of Home Gardens

Most gardens are not suited for the conservation of species with special requirements of area, soil, climate or habitat [215]. In fact, urban sprawl is one of the most threatening factors, by reducing natural habitats area and their ecological status, i.e. natural habitats conservation should be the priority [25]. Nevertheless, in the advent of an increasing urbanized and agriculture intensive world, wildlife gardening could create, within a small area, a diversity of microhabitats suited for several species [215]…..”

2.1. Health and Well-being, it refers again to sub-item 1.2. talking about the benefits you have with respect to well-being. 2.2 again describes the carbon footprint issue I mention in 1.3.

R10: We are specially thankful to the criticisms and doubts of the reviewer, that made us reinforce and clarify our division of the manuscript, namely by incorporating in the first sections (1.2, 1.3, 1.4) the advantages associated with health, food and biodiversity (with the global opinion in 1.5) while in the sections (2.1, 2.2, 2.3) we tried to discuss the drawbacks associated with the same issues, namely in the scope of the sustainable development goals (2.4 is the global opinion). When we had doubts (example Co2), issues could be tackled in both sections (1.2 and 2.1) Please see below the titles (and text associated with the revised sections):

1.5. Home Gardens Contribution to the Sustainable Development Goals

The authors define home gardens as gardens that might be characterized by their location, near or around a family house, for their (mostly) private use with a scope linked with the families’ conceptions and needs. We consider that their contribution to health and well-being can be separated within two major influences: nature exposure and out-door’ stimulation, renowned for its positive physiological and psychological benefits and the physical exercise associated with gardening, providing strength improvement, calories burning and, in general, a better physical and mental health (Table 3). Concerning diets, their role in boosting food diversity and nutrition should not be disregarded. We highlight that for low income and/or isolated regions, this is particularly relevant, including traditional medicine production (Table 3). The authors consider that the possibility of home gardens acting as carbon sink depends on several factors linked with management; more research is needed to enlighten this potential function (Table 3). Biodiversity conservation is a complex issue but, with careful management, design, correct size and habitats creation, home gardens matrix might contribute to sustain “wild” species metapopulations, by the plantation of native species (e.g. tree species) or by the resources associated with the “habitats” created by the gardener (Table 3).

Table 3. - Contribution of home gardening to the Sustainable Development Goals

Benefits

  • Nature exposure benefits to well-being
  • Outdoors stimulation and outdoors’ exercise benefits to health
  • Diet diversification, traditional medicines production and extra income
  • Carbon sequestration and biodiversity conservation contingent upon management
  1. Complementary Remarks on the Risks and Drawbacks of Home Gardens and Home Gardening in the scope of their Contribution to the Sustainable Development Goals

2.1. Risks for Health and Well-being of Home Gardens and Home Gardening

Musculoskeletal injuries are very common in individuals who farm the land [45], although addressing this type of injury in gardening is still very limited in the literature [49]. Some authors indicate that the causes for its manifestation are related with the presence of inadequate work practices (repetitive and performed for a prolonged period), bio-mechanical factors (improper handling of gardening tools, lifting and carrying heavy loads, repetitive flexion movements of the spine, excessive movement of the lumbar region or neck), use of poorly ergonomic tools, fatigue, and poor physical fitness [198-200]. Low back pain is very common and is aggravated by age, crop type, stress levels, and the presence of previous occupational injuries [200]. The manual and repetitive activities associated with gardening, such as planting, spraying, sweeping, and using shears can also lead to wrist and hand injuries. Short rest breaks, use of ergonomic tools, and elevated flowerbeds are some of the strategies that can help reducing the symptoms of fatigue and musculoskeletal discomfort associated with gardening. Other health risks associated with gardening involve exposure to chemical substances through skin contact, ingestion, or inhalation with dermatological, gastrointestinal, neurological, oncological, respiratory, and endocrine effects [201]. Elderly gardeners and immunosuppressed individuals are particularly sensitive to Legionellosis, an infectious disease caused by the exposure to Legionella bacteria, present in compost submitted to high temperatures [202]. Handling it might cause the release of microorganisms and bioaerosols, providing ideal conditions for the growth of fungi that lead to non-allergic, immunoallergic (rhinitis, allergic asthma) and inflammatory reactions [203]. Cuts and wounds resulting from handling thorny plants and power tools and gardening equipment facilitate the entry into the body of spores of the Clostridium tetani bacteria, resulting in the onset of muscle spasms, cramps, and even convulsions. Some plants and insects (bees, wasps, red ants) can also cause allergic reactions in some individuals. Tick bites, very common in gardens [204], affects the joints and nervous system [205]. Moistening dry compost before turning or using it, wearing gloves, keeping your arms covered when pruning plants likely to cause irritation may minimize the occurrence of some of these health risks. However, considering the current state of the art, more research is needed on this issue, namely by measuring exposure, deepening the underlying mechanisms, and demonstrating causality [206]. This is even more incomplete when it comes to home gardens and home gardening: research is needed to understand the real benefits of the spaces created and management practices on human health and well-being, something that ought to be done by integrated teams linking ecologists, health and social scientists with gardeners. With strong (and hopefully) positive results, policies might be developed to promote home gardens and home gardening (and gardens and gardening in general) in the scope of the one-health approach [206]. In fact, home gardens might contribute to halt habitat degradation/destruction and biodiversity loss, mitigate locally ongoing climate change and contribute to several human well-being and health benefits of experiencing nature.

2.2. Food Provision, Nutritional and Carbon Footprint Risks

Positive effects of home gardens and home gardening for food provision, diet diver-sity, nutrient supply and carbon footprint were highlighted in the previous section. For example, home gardens were pointed as relevant to obtain a continuous supply of daily foods for households in remote locations [207], supplementing diets with proteins, vita-mins, minerals and thus contributing for food security, food diversity and nutrition. However, drawbacks were reported: gardens are often located near roads or near intensive agricultural areas, which are more susceptible to be contaminated by heavy metals [208-210] and organic pollutants (for example: polycyclic aromatic hydrocarbons (PAHs)), antibiotics, and petroleum products and pesticides [209]. In this situation gardeners may be exposed to these substances, which are an important set of constraints pointed by several research studies [188]. In fact, growing foods within or near the main roads, factories or intensive agricultural fields, increases the chance of high concentration on the soil of potentially toxic elements such as As, Cr, Cu, Fe, Mn, Ni, Pb, Sn, and Zn, that can be an obstacle to produce safe and healthy fresh products [116-118]. Nonetheless, several solutions can be implemented to minimize the potential risk of soil contamination such as building raised beds for the crops, using amendments to stabilize contaminants in soil, adding thick layers of organic matter to the soil (i.e. providing a physical barrier to contamination), replacing contaminated soil with clean one or even using plant species that extract, degrade, contain or immobilize contaminants in soil [119]. Thus, despite the potential risk of contaminants in soils, several practices are currently available to (partially) overcome this problem. Anyway, whenever pesticides are used (insecticides, fungicides and herbicides) - gardeners often lack training in how to use them safely - harmful effects to human health could upsurge. Even if several “problematic” pesticides were banned from developed countries, in many other, they are still of common use. Moreover, a recent research revealed that in the UK, growers can easily purchase unauthorized pesticides online, including atrazine, an herbicide which has been banned for sale in the EU for more than a decade [115]. Another important criticism to home gardens and home gardening is linked with their food provision role: several authors suggest that only a small and mod-est contributions to overall food and nutritional needs are fulfilled [211]. Moreover, the majority of studies involving home gardening only address the potential of urban soil for food production and how much land or what types of soils would be required to feed the city’s population. In fact, comparisons are sometimes difficult to establish, since methods of crop production and types of crops differ among studies. The lack of data also complicates the comparisons between potential urban and rural home gardens and home gardening, but a question always raised is about the amount of food supplied by gardens to the households. In our point of view, the objectives of domestic and home gardens should not be to provide the complete needs of nutrients to households, but to complement, since extensive areas of farming already exist for this purpose. On the other hand, critics always refute the idea that gardens can effectively contribute for reducing the carbon footprint, mentioning that garden species need an extra-care with fertilization, watering and sanitary treatments [211]. The excessive use of mineral fertilizers, especially nitrogen and potassium might end in the groundwaters, but also the accumulation of pesticide residues in soils and foods, groundwater depletion, are among others, the problems raised [211]. Nonetheless, the majority of criticism and drawbacks reported, even opportune as reflection points, can be minimized using the correct “farming” practices. The use of organic fertilizers, green and organic amendments, natural substances or natural products for crop sanitary treatments, the rational use of irrigation water (only when necessary) have already prove could reduce the negative impacts in the environment [126]. The wide-spread use of mulching or compost, ground cover, vegetables and fruits in raised beds (filled with an uncontaminated soil), no tillage and sowing annual plants away from busy roads are practices to be considered [125]. All these practices can also contribute to the recovery of degraded soils in the gardens, but also capture different forms of atmospheric carbon, contributing in this way as a carbon sink [124].

2.3. Biodiversity and Nature Conservation: The Dark Side of Home Gardens

Most gardens are not suited for the conservation of species with special requirements of area, soil, climate or habitat [215]. In fact, urban sprawl is one of the most threatening factors, by reducing natural habitats area and their ecological status, i.e. natural habitats conservation should be the priority [25]. Nevertheless, in the advent of an increasing urbanized and agriculture intensive world, wildlife gardening could create, within a small area, a diversity of microhabitats suited for several species [215]. Conversely, several of the species selected by gardeners or attracted to live in our home gardens (e.g., cats; natural-ized and/or invasive species) may pose huge threats to our wild neighbors by spreading infectious diseases, predating several vertebrates and invertebrates but also compete for space, nutrients and light [140-142]. In fact, a relevant drawback related with home gar-dens and home gardening is linked with the chosen species, namely the potential of exotic species becoming invasive [216]. In fact, several plants and animals (and their associated parasites) brought to the garden in order to increase its beauty (e.g., colorful flowers) es-cape and become invasive in the wild habitats, creating immense challenges for conservationists by competing with native species, changing web links and fire regimes, and spreading new diseases [157]. Also, downfalls created by our urge for beauty, “cleanness” and pest control within our home garden heavens, by massively adding fertilizers and pesticides, might create traps and mortality events for several non-target species [143-145]. Again, gardener’s education and garden centers consulting should be a priority to tackle this problematic [217]. Another relevant aspect, namely in regions with water scarcity (e.g., Mediterranean region) is the preconception of “green” gardens (e.g., lawn) that need immense quantities of water (and chemicals) to maintain its features [218]. Apart from the costs associated with water consumption, this water is, many times, deviated from subterranean waters, wetlands or associated with the construction of reservoirs that further impact natural habitats [219]. Education again could pave the way for more sustainable gardens, by including novel irrigation techniques using grey waters, xeriscape concepts and mimicking regional natural habitats by choosing native species adapted to the local climate and soils [216].

2.4. Risks and Drawbacks of Home Gardens and Home Gardening to the implementation Sustainable Development Goals

Recognizing the potential interest of home gardens and home gardening for health and well-being, namely for elderly people, the authors also would like to stress that musculoskeletal injuries are a common problem, that could be reduced by specific gardening education directed to postural techniques and tools use, among other (Table 4). On the other hand, correct handling of chemicals (e.g. limiting the access to accredited gardeners) might prevent toxicological effects occurrence while proper clothing and hygiene can also minimize skin lesions from plants and arthropods and reduce infection by microorganisms. The authors recognize the risks of contamination of home gardens located near ur-ban/industrial or intensive agriculture areas with metals and/or pesticides (but also in some cases by the excessive use of agrochemicals by the gardener) that might end the legumes, fruits and groundwater (Table 4). Several techniques are available, from mulching to organic farming and cover crops, to help tackling the previous problematics, but more studies are needed to prove efficacy. In our opinion, these techniques might also reduce water consumption and contribute to capture carbon from the atmosphere. Biodiversity of home gardens might be enhanced by choosing the right species and correct management techniques, is by no means comparable to the biodiversity found on natural habitats (Table 4). Additionally, home gardens might impact significantly natural ecosystems functioning and biodiversity: the introduction of alien invasive species and water con-sumption in arid and semi-arid regions but again, environmental education of gardeners might make a difference (Table 4).

Table 4. – Risks and Drawbacks of Home Gardens and Home Gardening

Risks/Drawbacks

  • Musculoskeletal injuries
  • Ecotoxicological effects associated with agrochemicals
  • Infection by microorganisms of scars and skin lesions
  • Contamination by metals and other toxic substances
  • Invasive species introductions
  • Water consumption in arid regions”

2.3 the approach of the sub-theme is not clear with respect to what has been written, Biodiversity and Nature Conservation from where or from whom?

Of the orchards or of the environment. It reads a bit contradictory information.

R11: As stated in the previous response, we have changed the title and the section text, in order to fulfil our need to divide the advantages from the drawbacks, making the manuscript clearer to the reader. We hope that in the revised version, this question is clarify. Anyhow, we will be willing to add/change the text if the reviewer finds it incomplete. Please see below the new section 2.3

2.3. Biodiversity and Nature Conservation: The Dark Side of Home Gardens

Most gardens are not suited for the conservation of species with special requirements of area, soil, climate or habitat [215]. In fact, urban sprawl is one of the most threatening factors, by reducing natural habitats area and their ecological status, i.e. natural habitats conservation should be the priority [25]. Nevertheless, in the advent of an increasing urbanized and agriculture intensive world, wildlife gardening could create, within a small area, a diversity of microhabitats suited for several species [215]. Conversely, several of the species selected by gardeners or attracted to live in our home gardens (e.g., cats; naturalized and/or invasive species) may pose huge threats to our wild neighbors by spreading infectious diseases, predating several vertebrates and invertebrates but also compete for space, nutrients and light [140-142]. In fact, a relevant drawback related with home gar-dens and home gardening is linked with the chosen species, namely the potential of exotic species becoming invasive [216]. In fact, several plants and animals (and their associated parasites) brought to the garden in order to increase its beauty (e.g., colorful flowers) es-cape and become invasive in the wild habitats, creating immense challenges for conservationists by competing with native species, changing web links and fire regimes, and spreading new diseases [157]. Also, downfalls created by our urge for beauty, “cleanness” and pest control within our home garden heavens, by massively adding fertilizers and pesticides, might create traps and mortality events for several non-target species [143-145]. Again, gardener’s education and garden centers consulting should be a priority to tackle this problematic [217]. Another relevant aspect, namely in regions with water scarcity (e.g., Mediterranean region) is the preconception of “green” gardens (e.g., lawn) that need immense quantities of water (and chemicals) to maintain its features [218]. Apart from the costs associated with water consumption, this water is, many times, deviated from subterranean waters, wetlands or associated with the construction of reservoirs that further impact natural habitats [219]. Education again could pave the way for more sustainable gardens, by including novel irrigation techniques using grey waters, xeriscape concepts and mimicking regional natural habitats by choosing native species adapted to the local climate and soils [216].”

Final Remarks or conclusions should not be based on other works (References), rather, it must be written based on the results obtained and the objective of the manuscript.

R12: We have considered this suggestion particularly important, namely for a more assertive take-home message. In this way, a new short conclusion was introduced (and a reorganization of the discussion was made). The new conclusion:

“4. Conclusion

Our supported opinion aimed to describe and discuss the evidence of the effects of home gardens and home gardening on the well-being and health, nutrition, carbon foot-print reduction, biodiversity and nature conservation, fundamental issues for achieving the sustainable development goals (SDGs). The objectives were to understand its benefits (and drawbacks), provide an opinion reinforced by the literature, in order to guide scientists, managers and policy-makers in envisioning home gardens and home gardening as a humble but significant strategy in this scope. The strength of our supported opinion was its approach to understand the breadth of the authors’ opinion on the effects on selected SDGs.”

It is recommended to review wording.

R13: Dear reviewer, many thanks for this important aspect. All manuscript was revised searching for errors and grammatical issues. A version showing the changes made was uploaded.

Reviewer 3 Report (New Reviewer)

In this manuscript, the authors detail the contribution to the Sustainable Development Goals SDGs from the concept of home gardens and their impact on human well-being, human health, food security and biodiversity, and make recommendations. The logic is clear and the meaning is great. Even so, there are still some modifications need to be finished before it accepted. The following are the questions and some mistakes in this manuscript:

(1) Please make a clear distinction between garden and home garden, which are confused in the paper.

(2) The format of references cited in this paper is confused, please unify

(3) What's the special notation for line 156? Ps: (4.0 METs ? 70 kg ? (30 minutes)/(60 minutes))

(4) “CO2” in line 344 is wrong. Please change it to “CO2”.

(5) The horizontal lines following Table 1 and Table 2 are inconsistent.

(6) The format of the references is confused, please modify according to the journal format. Such as: year, DOI number, journal name, etc.

Author Response

Reviewer 3

In this manuscript, the authors detail the contribution to the Sustainable Development Goals SDGs from the concept of home gardens and their impact on human well-being, human health, food security and biodiversity, and make recommendations. The logic is clear and the meaning is great. Even so, there are still some modifications need to be finished before it accepted. The following are the questions and some mistakes in this manuscript:

R14: Dear reviewer, many thanks for your support and inspiring comments and suggestions.

1 Please make a clear distinction between garden and home garden, which are confused in the paper.

R15: Dear reviewer, many thanks for this important comment, that was considered very important so that the reader is able to understand the manuscript focus. Apart from a reinforcement in the title and subtitles of subsections and a more clear definition of home gardens, all manuscript was revised in order to check this issue, clarifying, whenever needed the home garden specificity or its connection with other types of gardens (some examples below):

1.1. Home garden definition and methodological framework

1.1.1. Home garden definition

A garden is a planned space, usually outdoors, set aside for the cultivation, display, and enjoyment of plants and other forms of nature [35]. Within gardens, home gardens are small areas (usually far below 1 hectare) surrounding the residential parcel, usually associated to family use (most home gardens are family gardens), characterized by combinations of various perennial and annual plants, sometimes in association with domestic animals and might include additional infrastructures such as ponds, greenhouses, green roofs, among other [35]. From pure aesthetic gardens to food production envisioned, sev-eral include mixed areas (e.g. agroforests) and diverse uses. Actually, home garden “ar-chitecture” and organization, the species’ chosen and the management options are linked with the local ecological conditions, but mostly with the options of the members of the household, providing a diverse and stable supply of services and benefits to the families [36,37]. Even if they are associated with urban domestic/family gardens and/or self-consumption, home gardens are also an important land use component in peri-urban and rural areas and in local food markets [38].”

2 The format of references cited in this paper is confused, please unify

R14: All references cited were verified. Many thanks for this important comment.

3 What's the special notation for line 156? Ps: (4.0 METs ? 70 kg ? (30 minutes)/(60 minutes))

R15: The notation was * (multiplication). Probably this occurred was transforming the manuscript from a word file to PDF. It was corrected in the revised version. Thank you very much!

4 “CO2” in line 344 is wrong. Please change it to “CO2”.

R16: We have changed in accordance with your suggestion. Again, many thanks!

5 The horizontal lines following Table 1 and Table 2 are inconsistent.

R17: The lines were formatted. Thanks again for this important detail.

6 The format of the references is confused, please modify according to the journal format. Such as: year, DOI number, journal name, etc.

R18: All references cited were verify and unified. Many thanks for this important comment.

Reviewer 4 Report (New Reviewer)

I read the article with great interest and I would only propose some minor changes and additions that could improve the manuscript; in detail:

 1.4. Implication of Gardens’ Structure and Management to Biodiversity

It is suggested to insert the proposed images directly in the text for a clearer and more immediate understanding.

2.1. Health and Well-being / 2.2. Food Provision, Nutritional Effects and Carbon Footprint / 2.3. Biodiversity and Nature Conservation

It is suggested to collect in the table the main positive and negative aspects identified and exposed for a greater exploitation of the results obtained.

Author Response

Reviewer 4

I read the article with great interest and I would only propose some minor changes and additions that could improve the manuscript; in detail:

R19: Dear reviewer, many thanks for your support and inspiring comments and suggestions.

1.4. Implication of Gardens’ Structure and Management to Biodiversity

It is suggested to insert the proposed images directly in the text for a clearer and more immediate understanding.

R20: Dear reviewer, many thanks for your suggestion. We have introduced the images directly in the main manuscript.

2.1. Health and Well-being / 2.2. Food Provision, Nutritional Effects and Carbon Footprint / 2.3. Biodiversity and Nature Conservation

It is suggested to collect in the table the main positive and negative aspects identified and exposed for a greater exploitation of the results obtained.

R21: Again, we would like to acknowledge your suggestion, that was considered fundamental to increase the readability of the manuscript. As we have reorganized the manuscript, with 1.2., 1.3 and 1.4 mostly dealing with advantages and 2.1, 2.2 and 2.3 with problems and drawbacks, two tables were introduced (table 3 and table), along with text changes associate with both parts (1.5 and 2.4 respectively):

1.5. Home Gardens Contribution to the Sustainable Development Goals

The authors define home gardens as gardens that might be characterized by their location, near or around a family house, for their (mostly) private use with a scope linked with the families’ conceptions and needs. We consider that their contribution to health and well-being can be separated within two major influences: nature exposure and out-door’ stimulation, renowned for its positive physiological and psychological benefits and the physical exercise associated with gardening, providing strength improvement, calories burning and, in general, a better physical and mental health (Table 3). Concerning diets, their role in boosting food diversity and nutrition should not be disregarded. We highlight that for low income and/or isolated regions, this is particularly relevant, including traditional medicine production (Table 3). The authors consider that the possibility of home gardens acting as carbon sink depends on several factors linked with management; more research is needed to enlighten this potential function (Table 3). Biodiversity conservation is a complex issue but, with careful management, design, correct size and habitats creation, home gardens matrix might contribute to sustain “wild” species metapopulations, by the plantation of native species (e.g. tree species) or by the resources associated with the “habitats” created by the gardener (Table 3).

Table 3. - Contribution of home gardening to the Sustainable Development Goals

Benefits

  • Nature exposure benefits to well-being
  • Outdoors stimulation and outdoors’ exercise benefits to health
  • Diet diversification, traditional medicines production and extra income
  • Carbon sequestration and biodiversity conservation contingent upon management

and

2.4. Risks and Drawbacks of Home Gardens and Home Gardening to the implementation Sustainable Development Goals

Recognizing the potential interest of home gardens and home gardening for health and well-being, namely for elderly people, the authors also would like to stress that musculoskeletal injuries are a common problem, that could be reduced by specific gardening education directed to postural techniques and tools use, among other (Table 4). On the other hand, correct handling of chemicals (e.g. limiting the access to accredited gardeners) might prevent toxicological effects occurrence while proper clothing and hygiene can also minimize skin lesions from plants and arthropods and reduce infection by microorganisms. The authors recognize the risks of contamination of home gardens located near ur-ban/industrial or intensive agriculture areas with metals and/or pesticides (but also in some cases by the excessive use of agrochemicals by the gardener) that might end the legumes, fruits and groundwater (Table 4). Several techniques are available, from mulching to organic farming and cover crops, to help tackling the previous problematics, but more studies are needed to prove efficacy. In our opinion, these techniques might also reduce water consumption and contribute to capture carbon from the atmosphere. Biodiversity of home gardens might be enhanced by choosing the right species and correct management techniques, is by no means comparable to the biodiversity found on natural habitats (Table 4). Additionally, home gardens might impact significantly natural ecosystems functioning and biodiversity: the introduction of alien invasive species and water con-sumption in arid and semi-arid regions but again, environmental education of gardeners might make a difference (Table 4).

Table 4. – Risks and Drawbacks of Home Gardens and Home Gardening

Risks/Drawbacks

  • Musculoskeletal injuries
  • Ecotoxicological effects associated with agrochemicals
  • Infection by microorganisms of scars and skin lesions
  • Contamination by metals and other toxic substances
  • Invasive species introductions
  • Water consumption in arid regions

Round 2

Reviewer 2 Report (New Reviewer)

No Comments

Author Response

Dear reviewer, many thanks for your suggestion. We have changed the manuscript sub-title to " Biodiversity and Nature Conservation: The Downside of Home Gardens".

With kind regards,

Mário Santos

This manuscript is a resubmission of an earlier submission. The following is a list of the peer review reports and author responses from that submission.

Round 1

Reviewer 1 Report

This Special Issue is "Environmental Sustainability of Agricultural Systems: Concepts, Practices and Drawbacks". This article starts with home gardening to explain the benefits of human disease, stress, and quality of life, and ends with agriculture-related issues. This article has a sense of tautology because it is unlikely that home gardening can help agricultural system around the world achieve sustainability.

First of all, this article does not fit into the category of the special issue "sustainable agricultural systems" at all, but it only discusses how home gardening can help human disease, stress, quality of life, and agriculture-related issues. Second, home gardening does not contribute to the sustainability of agricultural systems around the world. Authors are advised to submit to other more suitable journals in MDPI.

Reviewer 2 Report

The research methodology is unclear.

Figure 1 and 3 are unsuitable for a scientific journal of this level.

The article is based on the opinions of other authors. The opinion of the authors of this study is poorly expressed

The results of research by other authors are not recommended in the conclusions.

Reviewer 3 Report

Dear authors, 

I read your opinion carefully. I support your opinion that gardens and gardening is an activity which can lead to social and ecological benefits, and I can see you have been reading enough literature to support this opinion. I like you also add the risks, in especially the agricultural subsection. The risks in the health subsection can be expanded in my opinion, as this is lacks a bit this critical view. 

Major comments

1) I miss your definition of a garden in the introduction. What makes a garden a garden? Do you look to the performance of gardening? Or the practice? Or do the space that we perceive as garden? L65 hints it is about home gardens.  

When I read your opinion, it seems it is mostly about the practice? 

But you are also citing some research which is more about exposure to greenness (and is more about the space aspect). 

Also, some research you are using for your arguments might not be conducted in gardens as you imagined, like for example some of the *nature fix* research you are quoting in the part of health benefits.  

(I call this the nature fix research/discourse, because this is the research that thinks that exposure to nature will lead to positive benefits. However, many gardeners, who experience also this eco-socialisation dimension, might feel more stress because of eco-anxiety and have to find tools with that. Just to make a remark it is a complex relationship.)  

2) 

L606. It seems your opinion piece is a call for research on missing gaps. I am not sure if an opinion is the right aspect.nI did not learn anything new when I read subsection 1, and even 2. It is not very provocative. You did some mapping of literature (but you did not explain how you selected the literature) to show some gaps in research. So… why did you choose for a systematic mapping/ literature? 

3) As this is an opinion piece, and partly based on your own research and experiences, I would like a note about your background and a reflection on how your background might have influenced the way how you perceive urban/home gardens and that it is disregarded in the local translations of the sustainable development goals. I feel your call is based on some observations of neglection of gardens in mostly your country/region. Is that the case of the rest of the world? For example, for East-Europe? How representative is your opinion? For who is this opinion? Fellow researchers in Portugal and Brasil only? 

4) Perhaps you can write a few sentences about your methods, or a declaration on which previous research and experiences you are building further on/building your opinion. 

Minor comments

1) I was a bit confused if this was about gardens or urban gardens, because you talk at some point about cities in the introduction. To make more clear it is about urban gardens, by adding ‘urban’ before garden in the title and abstract

2)See abstract. Are gardens in post-war Europe merely for leisure? There is a whole body of research on (post-socialist) gardening in East-Europe where gardening is not done for leisure

3) Please check your English. Some sentence constructions felt awkward. 

4) Small remark about grass lawns. It is defended in your article, but in green discourse, there is a lot of resistance against grass lawns, often described as dead ecological zones. Moreover, I was reading about this new trend of robots that mow the grass and apparently kill hedgehogs at night. So this idea of a perfect always-cut grass lawn might lead to the decay of some species. What is your opinion about the drawback of this 'cleanness', or even need for control related to this trend of perfect cut-grass lawns?